# An assessment of vaccine wastage in the Solomon Islands

**Ibrahim Dadari** [ID]¹, **Laura Ropiti**², **Aven Patson**², **Philip Okia**², **Jenny Narasia**²,
**Timothy Hare'e**², **Salome Namohunu**¹, **Divinal Ogaoga**², **Jenny Gaiofa**², **Effua Usuf** [ID]³ *

**1** United Nations Children's Fund (UNICEF) Pacific, UN Joint Presence, ANZ Haus, Honiara, Solomon
Islands, **2** Ministry of Health and Medical Services, Honiara, Chinatown, Solomon Islands, **3** Medical
Research Council Unit The Gambia @London School of Hygiene and Tropical Medicine, Fajara, The Gambia

* eusuf@mrc.gm

Institute for Global Health India, INDIA

**Data Availability Statement:** All relevant data are
within the paper and its Supporting information
files.

**Funding:** This work was funded by Gavi the
Vaccine Alliance and implemented by UNICEF

## Abstract

Calculating vaccine wastage rates supports vaccine forecasting and prevents stock outs/
over-stock at central and immunisation delivery facilities. Ensuring there are sufficient vac-
cines on the several small islands of The Solomon Island while minimising waste is a chal-
lenge. Twenty-two health facilities were selected randomly from six purposefully identified
provinces in the Solomon Islands and across the different levels of the health service. Addi-
tional data were obtained from the national medical stores and the Expanded Programme
on Immunisation (EPI) monthly reports for 2017 and 2018. All the selected facilities were vis-
ited to observe stock management practices. We calculated wastage rates for each vaccine
antigen in the EPI and described the type of wastage. We found a wide variation in the aver-
age wastage rates at the second level medical stores which may be attributed to the partial
availability of wastage data. The overall wastage rate for 20-dose BCG was 38.9% (18.5–
59.3), 10-dose OPV was 33.6% (8.1–59.1), and single dose PCV was 4.5% (-4.4–13.5).
The data from the two smaller and farthest provinces were incomplete/not available and did
not contribute to the overall wastage rates. About 50% of the reported wasted doses at the
facility were reported as "damaged" vials. Wastage rates were high for the multidose vials
and slightly lower for the single dose vials which were also higher than the indicative rates.
There is a need to improve recording of vaccine wastage through continuous monitoring for
better forecasting and program effectiveness.

## Introduction

The global number of under-5 deaths was estimated as 5.3 million in 2018, a reduction by half
since 1990 [93 deaths per 1000 live births to 39 in 2018] [1]. More than half of these under-5
deaths are due to diseases that are preventable and treatable through simple, affordable inter-
ventions. Vaccination though, remains the most cost-effective health intervention. Vaccines
prevent 2–3 million deaths every year, and an additional 1.5 million will be saved if global cov-
erage of vaccines improves [2]. However, in 2019 about 20 million children were not

through grant code SC180268 and UNICEF Non-grant funds. Both authors (ID & EU) were also paid salary/consultancy using the Gavi funds with grant code SC180268. The funders had no role in the study design, data collection and analysis, decision to publish, or preparation of the manuscript.

**Competing interests:** The authors have declared that no competing interests exist.

vaccinated. WHO advocates for continuation of routine vaccination even in the light of the current Covid-19 pandemic [3].

Vaccine wastage is the sum of vaccines discarded, damaged destroyed or not accounted for [4]. Vaccine wastage is expected and was estimated to be about 50% globally in 2005, a more recent estimate is need [5]. Wastage maybe from 'closed vials' in relation to storage temperature, and transportation or from 'open vials' waste from multi dose vials that are discarded at the end of a session, spillage or breakage. Vaccine wastage may be unavoidable but needs to be within acceptable rates. For example, opening a multidose vial for a few children in a rural setting to avoid missed opportunities in a hard-to-reach area is more acceptable than wastage due to freezing or expiry. A good knowledge of the national wastage rates would guide and prevent stock outs and over stock, aid in the decision on the appropriate vaccine presentation, immunization size and supply chain infrastructure.

Over 50 countries including the Solomon Islands have benefitted from GAVI and other partners to national immunization. As countries graduate from GAVI, reducing wastage would contribute to the affordability and sustainability of the vaccination programmes. To support countries, WHO recently revised its Global Indicative wastage rates calculator with a new tool to support immunization programmes improve vaccine forecasting and the monitoring of the utilization of vaccines and wastage rates based on their service delivery settings [4].

The overall aim of this project was to quantify vaccine wastage in Solomon Islands and identify strategies to reduce wastage on the islands for better program efficiency. Vaccine wastage rates are not routinely estimated and monitored in the country and this study was the first documented attempt to estimate the actual vaccines wastage rates in the country.

## Methods

### Setting

The Solomon Islands are in the South Pacific Ocean lying to the east of Papua New Guinea and northwest of Vanuatu. There are more than 900 islands covering a total landmass of 28,400km$^2$. The country comprises the capital territory of Honiara and nine other provinces. The population was estimated at 653,248 in 2017, 50.5% of which are children and youth less than 19 years of age and the birth cohort was approximately 16,000 [6].

The climate is equatorial, and extremely humid throughout the year, with a mean temperature of 26.5˚C and few extremes of temperature throughout the year. The coolest period is between June and August. Rains are more frequent between November and April with occasional squalls or cyclones.

Expanded Programme on Immunization (EPI)—The immunization programme of the Solomon Islands was established in the early 1980s. Immunization of infants and pregnant mothers is carried out by nurses during child welfare clinics at fixed base facilities and outreach/mobile sessions. The programme offers the traditional vaccines (BCG, OPV, hepatitis B, measles, and Td) and the more recent new vaccines (e.g., PCV, HPV, and IPV) in various dose formulations i.e. single dose vials containing only one dose or multi-dose vials with more than one dose (Table A in S1 Text). Coverage in 2018 was more than 80% for all antigens except Hepatitis B birth dose which was 66% [7].

Medical supplies are managed through a three-tier system. At the central level, the National Medical Store (NMS) receives, processes, and stores all medical supplies including vaccines. All vaccines are procured through UNICEF and stored in the Central vaccine stores which is part of the NMS. The supplies are delivered to the 18 second level medical stores (SLMSs) which serve as the intermediate distribution centers at the provincial level, these include area health clinics (AHCs). There are about 350 health facilities at the lowest level of the supply

chain consisting of hospitals, AHCs, rural health centres (RHCs) and Nurse Aid Posts (NAPs). Delivery from the central stores to the SLMs and directly to isolated clinics is bi-monthly, by air or by boat for areas without flight access. From the SLMS, vaccines are delivered every month to the frontline clinics or weekly for the child welfare clinics in areas without functional active cold chain on site. Vaccines are stored in solar refrigerators which are found in most clinics across the country that have no mains power supply, or compression type ice-lined refrigerators mostly in the SLMs connected to mains power supply. SLMs also have freezers for storing ice-packs and vaccines needing freezing. Cold boxes and vaccines carriers are available.

## Sampling

The study was conducted in selected SLMS and other health facilities across the three levels of the medical supply chain. The national medical store, eight SLMS and 14 health facilities at the lowest level were included from six Provinces. The provinces were purposefully selected to include the two largest, two smallest and Honiara which hosts the national medical store. Within each Province, SLMS, AHCs and RCHs and NAPs were randomly selected with the total number proportionate to the number of facilities in the Province (Table B in S1 Text). One selected facility was not operational and was replaced at random with another in the same province.

## Sources of data

The data sources are stated in (Fig 1). At the facilities, we reviewed the vaccine stock control books, and observed practices at the day of data collection. We checked the readings on temperature chart on the freezer for July 2019, availability of policy documents, target population, frequency of immunization clinics and physically counted vaccine stock in the refrigerator.

From the Ministry of Health and Medical Services (MHMS) we additionally obtained electronic data on number of vaccinations administered at the facilities. Each facility completes a monthly (activity) health facility report form which is forwarded to the MHMS where the data are entered into an electronic database, Demographic Health Information Software 2 (DHIS2). The data on immunisations were collected from the DHIS2 as most facilities did not have a copy of their submitted forms.

The NMS has been using the m-Supply (by sustainable solutions New Zealand), an electronic inventory control system since 2005. We retrieved from the inventory, the number of vaccines supplied at national level and distributed to each facility. A mobile version of the -mSupply is being rolled out to the SLMSs and some primary health facilities.

**National medical store**
- m-Supply for stocks and distribution of vaccines

**SLMS**
- Vaccine Stock Control Books

**Health facility**
- Vaccine Stock Control Books
- Practice at the facility

**DHIS2**
- Immunisation records from monthly reports

**Fig 1. Sources of data for the assessment.**

### Data collection

We reviewed retrospective data for January through December for 2017 and 2018 and observed practices at the selected facilities at the time of data collection between July and August 2019. One health facility did not operate from April to October 2017.

Fifteen data collectors, including nurses and public health graduates were trained over two days. Day 1 focused on the overall objectives, methodology and data collection tools and day 2 was a practical session at two facilities that were not selected to be part of the assessment. We had six supervisors who were staff members of the EPI one for each province. The supervisors had an initial briefing meeting and took part in the training for the field staff.

### Data management and analysis

The data were entered in excel and exported to Stata version 15 for the analysis. Vaccine wastage rate, for each antigen in the EPI, was estimated at three levels: a) NMS, b) SLMS and c) other primary health facilities. At the national medical stores, we estimated the proportionate wastage rates as the number of doses discarded/ (start balance + number of doses received) x 100 [5].

Due to the lack of data at the SLMS the proportionate wastage rate was not estimated. The wastage rate was rather estimated based on the quantity of doses supplied to the facility and the number of immunisations doses issued. These rates may be misleading as all vaccines distributed from a store are considered as used which is never the case in real life [5].

Among primary level facilities AHCs, RHCs and NAPS wastage rate was estimated based on the recorded number of doses wasted at the facility and using the quantity of doses received from the SLMSs in relation to the number of recorded immunisations.

We calculated wastage rates at the intermediary and primary level by first estimating usage as follows:

1. Usage = immunised/(number of doses wasted + immunised)*100 and

2. Usage = (immunised/Doses supplied) *100.

The wastage rate was then 100—usage in each instance.

The type of vaccine wastage was described based on the available data at national and facility level. This project was approved by the Solomon Islands Health Research and Ethics Review Board.

## Results

### Vaccine supply chain

The National medical stores and 22 facilities in total were visited. The SLMS receive their vaccine supply from the NMS (Fig 2). Two facilities 603 (AHC) and 802 (RHC) also received vaccines directly from the NMS. The supply source for one facility in Honiara was uncertain. It appeared some were received directly from the NMS and others via an SLMS.

### Vaccine wastage

**a. National medical store.** The total number of vaccines received and distributed from the NMS are shown in Table 1.

The data were complete for all eight vaccines except OPV in 2017. There was a switch from TT to Td vaccine in 2018. Starting balance for OPV in 2018 was zero from the available records in the m-Supply. All the reported expired BCG doses in 2018 were accounted for in the

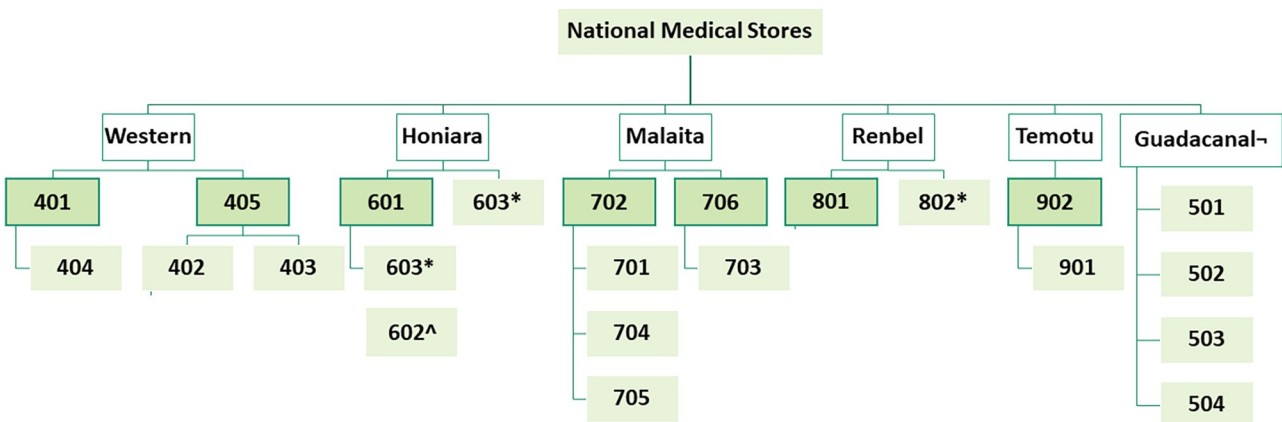

**Fig 2. Flow chart showing the selected facilities at the different levels of the health service.** Second level medical stores deeper green with borders receives supplies from NMS, others are NAP, RHC and AHC. *though not an SLMS also receives from NMS, for 603 it seems some supplies are received directly and others via the SLMS. ^ It is not certain where this facility receives it supplies, however, the facility supplies the mobile clinics. ¬ Facilities in the Guadalcanal province (GP) receive from the GP stores.

inventory. However, the MR and OPV expired doses did not account for all the negative inventory doses (Table 1).

The proportionate wastage for each vaccine in the EPI schedule for 2017 and 2018 is shown in Table 1. The wastage rates for the single dose per vial vaccines were higher in 2017 than 2018 with the highest being IPV, 8.5%. For the multidose vials, all had lower rates in 2018 except TT which had 73.5% wastage in 2018 compared to 9.0% in 2017. BCG recorded 24.1% in 2017 and zero percent wastage in 2018.

**b. Second level medical stores.** *Multidose vial vaccines.* From the data collected directly at the health facilities, discarded doses were reported for only three SLMSs (Table 2A). BCG had the highest wastage rates ranging from 91–95.7%. One province had wasted doses of OPV recorded but no record of the immunizations administered.

*Single dose vials.* No wasted doses were recorded for IPV and Penta during the two years. One facility recorded 31.7% wastage rate for PCV (Table 2B).

**c. Other health facilities (AHC RHC and NAP).** *Multidose vial vaccines.* BCG wastage rate ranged from 17.9–99.5%. Two provinces did not have records for wasted BCG doses for 2017 and 2018. (Table 3A). MR ranged from 1.6 to 75.8% from data for three provinces. Only two provinces had TT estimates of 4.1% and 27.6% both in 2018. OPV estimates were (44.3%) in 2018 and (53.4%) in 2017 from two provinces. Hepatitis B ranged from 9.3% to 47.4% (Table 3A).

*Single dose vials.* The lowest and highest rates 8.5%, and 64.8% for IPV and pentavalent vaccines (Table 3B).

Additional information is provided in Tables C and D in S1 Text on the vaccine wastage rates based on the reported discarded doses at the different categories of facilities.

## Overall wastage rates

The wastage rates by Province and overall are shown in Table 4. There were no wastage records from one Province, Renbel; the vaccine stock record books were not available at the two facilities visited. In Temotu, one facility recorded wasted doses but had no records of

**Table 1. Vaccine stock management national medical store, 2017 & 2018.**

| Vaccine | | Wastage rate (%) | Start dose | received | out | remaining | Inventory [c] | | Expired doses |
|---|---|---|---|---|---|---|---|---|---|
| | | | | | | | - ve | +ve | |
| **Multidose** [a] | | | | | | | | | |
| BCG | 2017 | 24.1 | 5978 | 18239 | 15715 | 8502 | 5839 | 5276 | 2427 |
| | 2018 | 0.0 | 8502 | 13952 | 10895 | 11559 | 1 | 102 | 0 |
| MR | 2017 | 20.1 | 6379 | 39401 | 30452 | 15328 | 9216 | 6095 | 0 |
| | 2018 | 18.8 | 15328 | 9463 | 16221 | 8570 | 4656 | 623 | 4358 |
| OPV [b] | 2017 | | | | | | | | |
| | 2018 | 28.0 | 0 | 29733 | 19053 | 10680 | 8315 | 12733 | 8315 |
| Td | 2018 | 0.0 | 0 | 10900 | 5220 | 5680 | 0 | 5324 | 0 |
| TT | 2017 | 9.0 | 6312 | 10104 | 9425 | 6991 | 1481 | 1104 | 0 |
| | 2018 | 73.3 | 6991 | 552 | 7543 | 0 | 5527 | 532 | 0 |
| HepB10 | 2017 | 75.0 | 16251 | 4937 | 19102 | 2086 | 15890 | 437 | 0 |
| | 2018 | 34.2 | 2086 | 2463 | 3139 | 1410 | 1556 | 1934 | 6 |
| **Single dose** | | | | | | | | | |
| HepB_S | 2018 | 0.2 | 8107 | 21222 | 17549 | 11780 | 61 | 562 | 0 |
| IPV | 2017 | 8.5 | 10318 | 48408 | 43793 | 14933 | 4990 | 0 | 0 |
| | 2018 | 1.8 | 14933 | 31111 | 30834 | 15210 | 841 | 511 | 0 |
| PCV | 2017 | 1.3 | 2637 | 92135 | 94560 | 212 | 1278 | 2236 | 0 |
| | 2018 | 0.1 | 212 | 102204 | 96976 | 5440 | 112 | 1004 | 0 |
| Penta | 2017 | 6.0 | 15965 | 147218 | 128244 | 34939 | 9860 | 18157 | 0 |
| | 2018 | 2.7 | 34939 | 115071 | 110028 | 39982 | 4037 | 1182 | 0 |

[a] all 10 dose vial except BCG which is a 20 dose vial

[b] No data for 2017

[c] Inventory- The inventory covers physical count at the stores; (+ve) are surplus doses and (-ve) are unaccounted for doses. These negative numbers recorded following stock inventory at NMS were included as wasted/discarded/missing vaccines in wastage rate estimation.

immunizations, and the second facility visited had no records at all. Western province had no wasted doses recorded for all vaccines in each of the five selected facilities in that region.

Overall wastage for BCG was 38.9% (18.5–59.3). The pooled wastage rate for PCV was 4.5% (-4.4–13.5) compared to 20.6% (-6.9–48.1) for pentavalent vaccine.

## Type of wastage

**National medical stores.** Only expired wasted vaccines were documented at the NMS, and these were not captured in the m-Supply as the system is not setup to record vaccine wastage as yet. The number of expired doses for BCG, MR, OPV, and Hepatitis B for the two years was obtained directly from the NMS. The inventory, which was conducted on a regular basis, showed differences/surplus in the stocks (Table 1).

**SLMS.** Fig 3 shows the proportion of vaccine wastage across all facilities. Two SLMS reported expired doses, OPV (quantity not stated) and Hepatitis B, 30 vials. In some facilities, wastage was reported but the type not specific, some may be unavoidable opened vial wastage e.g., 315 cumulative doses of MR over the duration of the two years in one facility (Table E in S1 Text).

**Other facilities.** From the vaccine records at the facilities, the type of wastage was documented for four service level facilities (Table E in S1 Text). The highest records of wastage at these facilities were 410 damaged MR vials, 380 vaccine vial monitor (VVM) stages 3 or 4, and

**Table 2. Wastage rate at the SLMS (A) Multidose vials and (B)single dose vials.**

(A) Multidose vials

| Vaccine | Province | Year | Wastage rate | Doses wasted | Immunised | Usage |
|---|---|---|---|---|---|---|
| BCG | Western | 2017 | 0.0 | 0 | 748 | 100.0 |
| | | 2018 | 0.0 | 0 | 778 | 100.0 |
| | Guadalcanal | 2017 | 0.0 | 0 | 643 | 100.0 |
| | | 2018 | 0.0 | 0 | 657 | 100.0 |
| | Honiara | 2017 | 95.7 | 264 | 12 | 4.3 |
| | | 2018 | 91.0 | 122 | 12 | 9.0 |
| | Malaita | 2017 | 0.0 | 0 | 923 | 100.0 |
| | | 2018 | 0.0 | 0 | 1172 | 100.0 |
| | Renbel | 2017 | 0.0 | 0 | 13 | 100.0 |
| | | 2018 | 0.0 | 0 | 13 | 100.0 |
| | Temotu | 2017 | 100.0 | 206 | 0 | 0.0 |
| | | 2018 | 100.0 | 84 | 0 | 0.0 |
| HepB | Western | 2017 | 0.0 | 0 | 776 | 100.0 |
| | | 2018 | 0.0 | 0 | 781 | 100.0 |
| | Guadalcanal | 2017 | 0.0 | 0 | 595 | 100.0 |
| | | 2018 | 0.0 | 0 | 663 | 100.0 |
| | Honiara | 2017 | 0.0 | 0 | 1 | 100.0 |
| | | 2018 | 62.5 | 10 | 6 | 37.5 |
| | Malaita | 2017 | 21.7 | 300 | 1082 | 78.3 |
| | | 2018 | 0.0 | 0 | 1108 | 100.0 |
| | Renbel | 2017 | 0.0 | 0 | 8 | 100.0 |
| | | 2018 | 0.0 | 0 | 13 | 100.0 |
| | Temotu | 2017 | 100.0 | 5 | 0 | 0.0 |
| | | 2018 | nd | 0 | 0 | nd |
| MR | Western | 2017 | 0.0 | 0 | 43 | 100.0 |
| | | 2018 | 0.0 | 0 | 258 | 100.0 |
| | Guadalcanal | 2017 | 0.0 | 0 | 523 | 100.0 |
| | | 2018 | 0.0 | 0 | 496 | 100.0 |
| | Honiara | 2017 | 0.0 | 0 | 468 | 100.0 |
| | | 2018 | 4.9 | 23 | 444 | 95.1 |
| | Malaita | 2017 | 0.0 | 0 | 179 | 100.0 |
| | | 2018 | 0.0 | 0 | 319 | 100.0 |
| | Renbel | 2017 | 0.0 | 0 | 22 | 100.0 |
| | | 2018 | 0.0 | 0 | 91 | 100.0 |
| | Temotu | 2017 | nd | 0 | 0 | nd |
| | | 2018 | nd | 0 | 0 | nd |
| OPV | Western | 2017 | 0.0 | 0 | 15 | 100.0 |
| | | 2018 | 0.0 | 0 | 720 | 100.0 |
| | Guadalcanal | 2017 | 0.0 | 0 | 51 | 100.0 |
| | | 2018 | 0.0 | 0 | 1670 | 100.0 |
| | Honiara | 2017 | 100.0 | 154 | 0 | 0.0 |
| | | 2018 | 0.0 | 0 | 1250 | 100.0 |
| | Malaita | 2017 | 0.0 | 0 | 424 | 100.0 |
| | | 2018 | 0.0 | 0 | 544 | 100.0 |
| | Renbel | 2017 | nd | 0 | 0 | nd |
| | | 2018 | 0.0 | 0 | 203 | 100.0 |
| | Temotu | 2017 | 100.0 | 1 | 0 | 0.0 |
| | | 2018 | nd | 0 | 0 | nd |

(*Continued*)

**Table 2.** (Continued)

| | | | | | | |
|---|---|---|---|---|---|---|
| TT | Western | 2017 | 0.0 | 0 | 109 | 100.0 |
| | | 2018 | 0.0 | 0 | 502 | 100.0 |
| | Guadalcanal | 2017 | 0.0 | 0 | 145 | 100.0 |
| | | 2018 | 0.0 | 0 | 1290 | 100.0 |
| | Honiara | 2017 | 0.0 | 0 | 247 | 100.0 |
| | | 2018 | 0.0 | 0 | 738 | 100.0 |
| | Malaita | 2017 | 0.0 | 0 | 724 | 100.0 |
| | | 2018 | 0.0 | 0 | 593 | 100.0 |
| | Renbel | 2017 | 0.0 | 0 | 17 | 100.0 |
| | | 2018 | 0.0 | 0 | 134 | 100.0 |
| | Temotu | 2017 | 0.0 | 0 | 117 | 100.0 |
| | | 2018 | 0.0 | 0 | 127 | 100.0 |

(B) Single dose vials

| Vaccine | Province | Year | Wastage rate | Doses wasted | Immunised | Usage |
|---|---|---|---|---|---|---|
| IPV | Western | 2017 | 0.0 | 0 | 65 | 100.0 |
| | | 2018 | 0.0 | 0 | 93 | 100.0 |
| | Guadalcanal | 2017 | 0.0 | 0 | 470 | 100.0 |
| | | 2018 | 0.0 | 0 | 469 | 100.0 |
| | Honiara | 2017 | 0.0 | 0 | 451 | 100.0 |
| | | 2018 | 0.0 | 0 | 351 | 100.0 |
| | Malaita | 2017 | 0.0 | 0 | 182 | 100.0 |
| | | 2018 | 0.0 | 0 | 146 | 100.0 |
| | Renbel | 2017 | 0.0 | 0 | 31 | 100.0 |
| | | 2018 | 0.0 | 0 | 55 | 100.0 |
| | Temotu | 2017 | nd | 0 | 0 | nd |
| | | 2018 | 100.0 | 4 | 0 | 0.0 |
| PCV | Western | 2017 | nd | 0 | 0 | nd |
| | | 2018 | 0.0 | 0 | 305 | 100.0 |
| | Guadalcanal | 2017 | nd | 0 | 0 | nd |
| | | 2018 | 0.0 | 0 | 1198 | 100.0 |
| | Honiara | 2017 | nd | 0 | 0 | nd |
| | | 2018 | 0.0 | 0 | 1042 | 100.0 |
| | Malaita | 2017 | nd | 0 | 0 | nd |
| | | 2018 | 31.7 | 220 | 475 | 68.3 |
| | Renbel | 2017 | nd | 0 | 0 | nd |
| | | 2018 | 0.0 | 0 | 143 | 100.0 |
| | Temotu | 2017 | nd | 0 | 0 | nd |
| | | 2018 | nd | 0 | 0 | nd |
| Penta | Western | 2017 | nd | 0 | 0 | nd |
| | | 2018 | 0.0 | 0 | 283 | 100.0 |
| | Guadalcanal | 2017 | | 0 | 0 | |
| | | 2018 | 0.0 | 0 | 1200 | 100.0 |
| | Honiara | 2017 | nd | 0 | 0 | nd |
| | | 2018 | 0.0 | 0 | 1026 | 100.0 |
| | Malaita | 2017 | nd | 0 | 0 | nd |
| | | 2018 | 0.0 | 0 | 475 | 100.0 |
| | Renbel | 2017 | nd | 0 | 0 | nd |
| | | 2018 | 0.0 | 0 | 140 | 100.0 |
| | Temotu | 2017 | nd | 0 | 0 | nd |
| | | 2018 | 100.0 | 9 | 0 | 0.0 |

**Table 3. Wastage rates for other facilities** (A) multidose vials and (B) single dose vials.

(A). Multidose vials

| Vaccine | Province | Year | Wastage rate | Doses discarded | Immunised | Usage |
|---|---|---|---|---|---|---|
| BCG | Western | 2017 | 0.0 | 0 | 25 | 100.0 |
| | | 2018 | 0.0 | 0 | 50 | 100.0 |
| | Guadalcanal | 2017 | 54.2 | 115 | 97 | 45.8 |
| | | 2018 | 17.9 | 14 | 64 | 82.1 |
| | Honiara | 2017 | na | 0 | 0 | na |
| | | 2018 | 100.0 | 27 | 0 | 0.0 |
| | Malaita | 2017 | 63.5 | 404 | 232 | 36.5 |
| | | 2018 | 16.3 | 32 | 164 | 83.7 |
| | Renbel | 2017 | 0.0 | nr | 2 | 100.0 |
| | | 2018 | 0.0 | nr | 1 | 100.0 |
| | Temotu | 2017 | 0.0 | 0 | 1 | 100.0 |
| | | 2018 | nd | 0 | 0 | nd |
| HepB | Western | 2017 | 0.0 | 0 | 22 | 100.0 |
| | | 2018 | 0.0 | 0 | 40 | 100.0 |
| | Guadalcanal | 2017 | 47.4 | 46 | 51 | 52.6 |
| | | 2018 | 40.4 | 42 | 62 | 59.6 |
| | Honiara | 2017 | nd | 0 | 0 | nd |
| | | 2018 | nd | 0 | 0 | nd |
| | Malaita | 2017 | 9.3 | 19 | 186 | 90.7 |
| | | 2018 | 0.0 | 0 | 134 | 100.0 |
| | Renbel | 2017 | nd | nr | nr | nd |
| | | 2018 | nd | nr | nr | nd |
| | Temotu | 2017 | nd | 0 | 0 | nd |
| | | 2018 | nd | 0 | 0 | nd |
| MR | Western | 2017 | 0.0 | 0 | 74 | 100.0 |
| | | 2018 | 0.0 | 0 | 235 | 100.0 |
| | Guadalcanal | 2017 | 35.7 | 81 | 146 | 64.3 |
| | | 2018 | 75.8 | 414 | 132 | 24.2 |
| | Honiara | 2017 | 0.0 | 0 | 29 | 100.0 |
| | | 2018 | 43.8 | 142 | 182 | 56.2 |
| | Malaita | 2017 | 13.0 | 41 | 275 | 87.0 |
| | | 2018 | 1.6 | 15 | 943 | 98.4 |
| | Renbel | 2017 | 0.0 | nr | 21 | 100.0 |
| | | 2018 | 0.0 | nr | 8 | 100.0 |
| | Temotu | 2017 | 0.0 | 0 | 2 | 100.0 |
| | | 2018 | 0.0 | 0 | 3 | 100.0 |
| OPV | Western | 2017 | | 0 | 0 | |
| | | 2018 | 0.0 | 0 | 365 | 100.0 |
| | Guadalcanal | 2017 | 100.0 | 158 | 0 | 0.0 |
| | | 2018 | 0.0 | 0 | 281 | 100.0 |
| | Honiara | 2017 | 0.0 | 0 | 655 | 100.0 |
| | | 2018 | 44.3 | 808 | 1017 | 55.7 |
| | Malaita | 2017 | 53.4 | 303 | 264 | 46.6 |
| | | 2018 | 0.0 | 0 | 1168 | 100.0 |
| | Renbel | 2017 | 0.0 | 0 | 5 | 100.0 |
| | | 2018 | 0.0 | 0 | 16 | 100.0 |
| | Temotu | 2017 | nd | 0 | 0 | nd |
| | | 2018 | nd | 0 | 0 | nd |

(*Continued*)

**Table 3.** (Continued)

| | | Year | Wastage rate | Doses discarded | Immunised | Usage |
|---|---|---|---|---|---|---|
| **TT** | Western | 2017 | 0.0 | 0 | 4 | 100.0 |
| | | 2018 | 0.0 | 0 | 144 | 100.0 |
| | Guadalcanal | 2017 | 0.0 | 0 | 179 | 100.0 |
| | | 2018 | 4.1 | 14 | 327 | 95.9 |
| | Honiara | 2017 | 0.0 | 0 | 990 | 100.0 |
| | | 2018 | 27.6 | 352 | 922 | 72.4 |
| | Malaita | 2017 | 0.0 | 0 | 542 | 100.0 |
| | | 2018 | 0.0 | 0 | 656 | 100.0 |
| | Renbel | 2017 | 0.0 | 0 | 27 | 100.0 |
| | | 2018 | 0.0 | 0 | 20 | 100.0 |
| | Temotu | 2017 | 0.0 | 0 | 1 | 100.0 |
| | | 2018 | nd | 0 | 0 | nd |

(B) Single dose vials

| Vaccine | Province | Year | Wastage rate | Doses discarded | Immunised | Usage |
|---|---|---|---|---|---|---|
| **IPV** | Western | 2017 | 0.0 | 0 | 55 | 100.0 |
| | | 2018 | 0.0 | 0 | 87 | 100.0 |
| | Guadalcanal | 2017 | 8.5 | 12 | 129 | 91.5 |
| | | 2018 | 20.8 | 21 | 80 | 79.2 |
| | Honiara | 2017 | 0.0 | 0 | 1 | 100.0 |
| | | 2018 | 0.0 | 0 | 6 | 100.0 |
| | Malaita | 2017 | 25.3 | 65 | 192 | 74.7 |
| | | 2018 | 0.0 | 0 | 245 | 100.0 |
| | Renbel | 2017 | 0.0 | 0 | 14 | 100.0 |
| | | 2018 | 0.0 | 0 | 7 | 100.0 |
| | Temotu | 2017 | nd | 0 | 0 | nd |
| | | 2018 | nd | 0 | 0 | nd |
| **PCV** | Western | 2017 | nd | 0 | 0 | nd |
| | | 2018 | 0.0 | 0 | 292 | 100.0 |
| | Guadalcanal | 2017 | nd | 0 | 0 | nd |
| | | 2018 | 0.0 | 0 | 250 | 100.0 |
| | Honiara | 2017 | nd | 0 | 0 | nd |
| | | 2018 | 0.0 | 0 | 20 | 100.0 |
| | Malaita | 2017 | 0.0 | 0 | 150 | 100.0 |
| | | 2018 | 0.0 | 0 | 894 | 100.0 |
| | Renbel | 2017 | nd | 0 | 0 | nd |
| | | 2018 | 0.0 | 0 | 13 | 100.0 |
| | Temotu | 2017 | 0.0 | 0 | 9 | 100.0 |
| | | 2018 | nd | 0 | 0 | nd |
| **Penta** | Western | 2017 | nd | 0 | 0 | nd |
| | | 2018 | 0.0 | 0 | 276 | 100.0 |
| | Guadalcanal | 2017 | nd | 0 | 0 | nd |
| | | 2018 | 0.0 | 0 | 264 | 100.0 |
| | Honiara | 2017 | nd | 0 | 0 | nd |
| | | 2018 | 64.8 | 35 | 19 | 35.2 |
| | Malaita | 2017 | 0.0 | 0 | 149 | 100.0 |
| | | 2018 | 0.0 | 0 | 855 | 100.0 |
| | Renbel | 2017 | nd | 0 | 0 | nd |
| | | 2018 | 0.0 | 0 | 13 | 100.0 |
| | Temotu | 2017 | 0.0 | 0 | 9 | 100.0 |
| | | 2018 | nd | 0 | 0 | nd |

nd—not done, nr—not recorded.

**Table 4. Average vaccine wastage rate among all facilities overall and by province.**

| Vaccine | Overall | Average wastage rates | | | | | |
|---|---|---|---|---|---|---|---|
| | | Province | | | | | |
| | | Western[a] | Guadalcanal | Honiara | Malaita | Renbel | Temotu[b] |
| BCG | 38.9 (18.5–59.3) | 0.0 | 26.0(-6.4–58.3) | 96.6(90.2–100.9) | 24.5(-11.1–60.0) | nr | 100 |
| HepB | 21.7(6.5–36.8) | 0.0 | 26.2(0.7–51.8) | 31.3(-33.7–96.2) | 9.6(-2.5–21.8) | nr | 100 |
| OPV | 33.6(8.1–59.1) | 0.0 | 30.0(-20.7–60.7) | 48.1(-12.1–108.3) | 30.7(-32.3–98.7) | nr | 100 |
| Penta | 20.6(-6.9–48.1) | 0.0 | 0.0 | 32.4(-35.0–99.8) | 0.0 | nr | 100 |
| PCV | 4.5 (-4.4–13.5) | 0.0 | 0.0 | 0.0 | 15.8(-16.6–48.3) | nr | 100 |
| MR | 20.4(2.8–38.1) | 0.0 | 35.6(-9.1–76.2) | 16.3(-12.5–45.1) | 14.9(-12.6–40.5) | nr | 100 |
| TT | 2.5(-1.1–6.0) | 0.0 | 2.4(-2.5–7.4) | 6.9(-7.5–21.3) | 0.0 | nr | 0.0 |
| IPV | 16.8(-0.7–34.2) | 0.0 | 15.6(-8.5–39.7) | 0.0 | 18.5(-19.5–56.5) | nr | 100 |

Wastage rate average using doses wasted and immunisations.

[a] immunizations recorded but no wasted doses recorded

[b] 100% -wasted doses recorded, but no immunizations recorded, 0% -no wasted doses recorded but no immunizations

Nr not recorded.

360 damaged PCV vials at three different facilities, respectively. Of the 4111 records of VVM status in the record books (all facilities), only 1% were VVM3/VVM4.

## Vaccine stock management at the facility level

Table 5 shows the observations at the facilities. We found that there was no refrigerator in one facility at the time of the survey, one had no refrigerator during the period under review 2017/18 (they had just received a new refrigerator from UNICEF), and in a third facility, there were no vaccines in the fridge.

Vaccine target population estimates were available at seven facilities. Four of them had targets for 2019, two still had 2018 targets and, one had 2017 targets visibly displayed.

Forty percent (9/22) had weekly fixed site immunization clinics and 88.9% (16/18) had regular satellite or outreach clinics. Fifty percent (10/20) received vaccine supply monthly and the majority 70% (14/20) request for their vaccines from the SLMs/NMS. Sixty-five percent (13/20) use the target population to determine their vaccine needs.

Sixty percent (12/20) reported vaccine stock outs in the previous six months. Sixty percent (12/20) return vaccines to the cold chain as per the MDVP. Forty percent (8/20) had supervisory visits in the three months preceding the survey.

## Vaccines—Physical count at the facilities

All facilities had pentavalent vaccines, though one had only one vial remaining. There were stock outs of various vaccines at the time of physical count (Table 6). There were no expired doses of PCV, Penta, IPV and TT. All other vaccines had expired doses in one or more facility. Seven facilities had Td that will expire in Sep/Nov 2019. All Hepatitis B, OPV and Td open vials in the refrigerators were not labelled. One facility had 32 doses of IPV with VVM 3 in the refrigerator. Only four facilities (20%) had up to date records of their stock in the record books. This was confirmed during physical counting at the time of the visit.

## Discussions

This project set out to quantify for the first time, vaccine wastage in the Solomon Islands, known for its small and sparse population and logistical difficulties of reaching communities

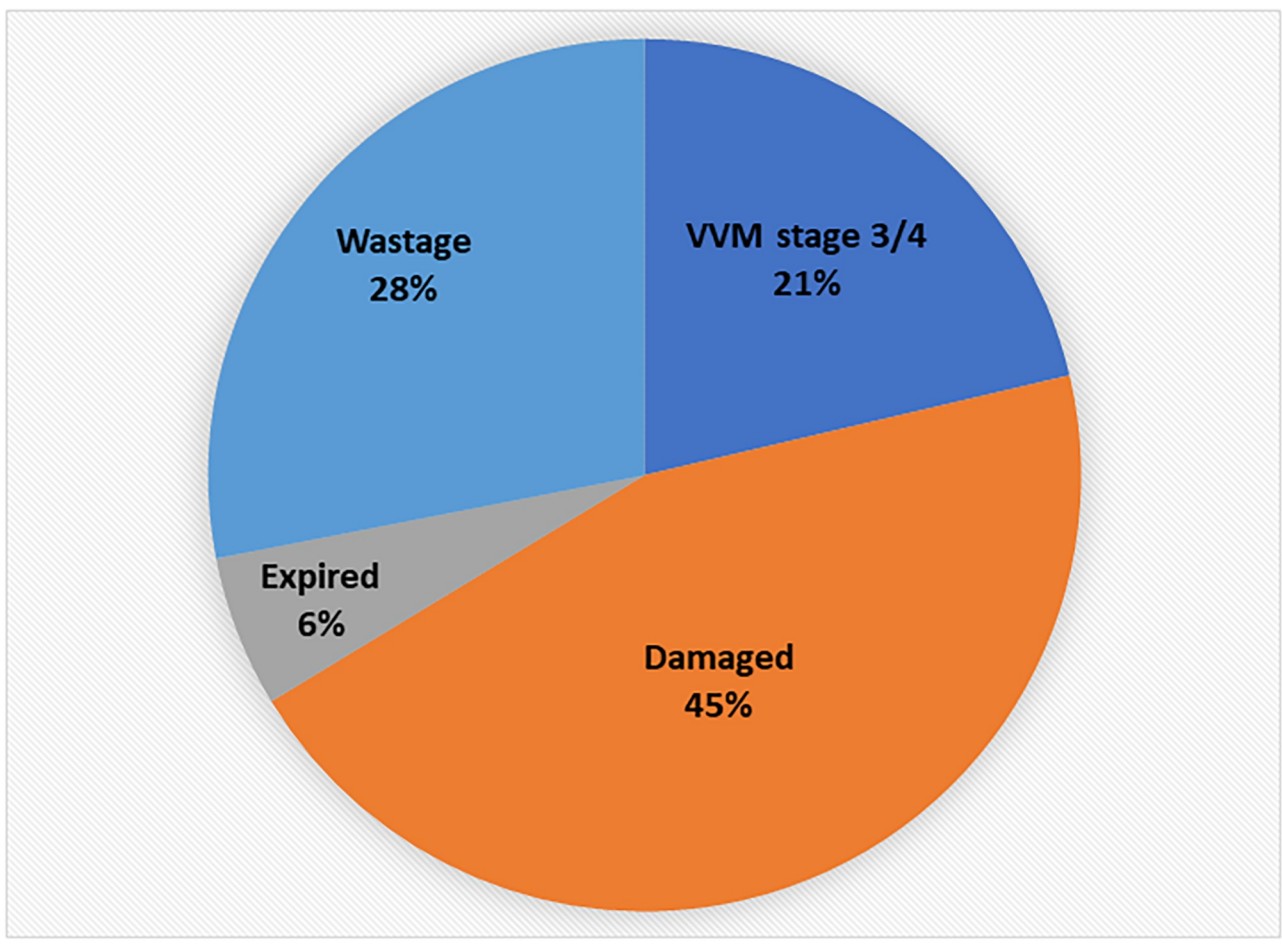

**Fig 3. Showing types of wastage at the facilities**\*. \*Power outage/off and refrigeration working function well were reported but the number of doses discarded for these reasons was not stated and hence not reflected in the proportions on the chart. Wastage/damaged were not defined further in the records. VVM—Vaccine vial monitor is a label containing a heat-sensitive material which is placed on a vaccine vial to register cumulative heat exposure over time.

in the different Islands. In general, we found that the overall wastage rates for all vaccines were within the WHO recommended rates, however there were huge gaps in the data particularly at the service delivery level.

During the two years under review, the proportionate wastage rates at the central vaccines stores were all within acceptable ranges for all but two vaccines, Hepatitis B in 10 dose vial and TT. The country had hepatitis B in two formulations, 10 dose and single dose vials and in 2018, the country switched from TT to Td vaccines, which occurred in a few other Pacific Islands countries as well. These transitions may have affected the wastage rates for these specific antigens in one way or another.

Wastage doses were rarely recorded at the SLMS. One facility had high rates >90% for BCG both in 2017 and 2018. These are among the highest rates recorded in LMIC [8–10]. Wastage of BCG has been shown to be related to the size of the immunisation session such that smaller sessions have larger wasted doses [11]. We did not assess the size of the immunisation clinics, but several facilities recorded less than 10 doses of BCG for a whole year. More complete data are needed to understand the extent of the differences.

**Table 5. Findings from the facility visits.**

| Activity/ Observation | Number of facilities (%) |
|---|---|
| *Target population at the facility* [a] | |
| 2019 | 4(18.1) |
| 2018 | 2(9.1) |
| 2017 | 1(4.5) |
| Not seen /unknown | 15 (68.2) |
| *Frequency of fixed site clinics* | |
| Weekly | 9(40.1) |
| Monthly | 4 (18.2) |
| Twice a week | 3(13.6) |
| Three times a week | 2(9.0) |
| Depends on the SLMS | 1 (4.5) |
| Unknown/Missing | 3(13.6) |
| *Satellite clinics conducted* | |
| Yes | 16(72.7) |
| No | 2(9.0) |
| Unknown | 4(18.2) |
| *Frequency of satellite clinics* | |
| Weekly | 4 (20.0) |
| Twice weekly | 3 (15.0) |
| Monthly | 3 (15.0) |
| Four times a month | 2 (10.0) |
| Others | 6 (30.0) |
| Policy/ guidelines on vaccine wastage available at the facility | |
| Yes | 7 (35.0) |
| No | 8 (40.0) |
| Yes, Not seen | 5 (25.0) |
| Supervisory visits in the last three months | |
| Yes | 8(40.0) |
| Temperature recording twice daily for July 2019 | 7(31.8) |
| Twice for some days | 4(18.2) |
| Once daily or some days | 2(9.0) |
| Not recorded | 8(36.4) |
| Twice all days except weekends | 1(4.5) |
| Frequency of vaccine supply to facility | |
| Monthly | 10 (45.5) |
| Every two months | 3(13.6) |
| Every three months | 1(4.5) |
| Weekly | 3(13.6) |
| Depends on the stock | 2(9.0) |
| Unknown/Missing | 2(9.0) |
| How are vaccines requested for | |
| Request for their vaccines | 14(70.0) |
| Based on estimates from SLMs | 6(30.0) |
| How are vaccine needs determined [a] | |
| No of doses used | 8)40.0) |
| Target population | 13(65.0) |
| Last vaccine supply | 1(5.0) |
| Child welfare registration | 1(5.0) |

(*Continued*)

**Table 5.** (Continued)

| Activity/ Observation | Number of facilities (%) |
|---|---|
| Reported stock out of vaccines in six months prior to visit Yes | 12(60.0) |
| BCG | 2 (10.0) |
| Hep B | 2(10.0) |
| Penta | 2 (10.0) |
| PCV | 3(15.0) |
| MR | 3(15.0) |
| TT | 2(10.0) |
| IPV | 3(15.0) |
| Wastage affected by | |
| Supply | 11(55.0) |
| Storage space | 7(35.0) |
| Cold chain failure | 8(40.0) |
| Vaccines returned to cold chain [b] | |
| Yes | 12(40.0) |
| BCG | 9(45.0) |
| Hep B | 4(20.0) |
| OPV | 9(45.0) |
| TT | 11(55.0) |
| Wastage calculated yes | 8(40.0) |
| No fridge | 1(4.5) |
| No vaccines in fridge | 1(4.5) |
| Nurse in charge not available | 2(9.0) |
| Last Medicine stock reported | 4(18.1) |

[a] the facilities displayed the target population for the year under indicated i.e. one facility had 2017 targets displayed on the wall

[b] refers to vaccines for which the multidose vial policy applies i.e those that can be reused up to 28 days after opening if appropriate conditions apply.

In general, for vaccines that the multidose vial policy (MDVP) covers, rates of less than 15–25% are considered acceptable. These indicative rates do not take country context into account, and hence the need for the recent WHO guidance for estimation of more accurate wastage rates [4]. We saw quite substantial wastage rates for OPV and TT among the few facilities with records. We noted also that all four facilities did not have the opened doses in the refrigerator labelled therefore determining the date it was opened may be a challenged and vaccines may end up being discarded. This contravenes requirements for the MDVP as specified in the Solomon Islands Vaccine Policy and global guidelines [12]. Introduction of the MDVP has been shown to be effective in reducing wastage for vaccines for which the policy applies [13].

At the other service level facilities, the wastage rates were consistent with findings in other places in developing countries [8, 10]. While overall PCV seems to be within acceptable ranges, wastage from the pentavalent vaccine which is administered similarly i.e., three doses at 2, 3 and 4 months were much higher. These figures are from only two provinces and should be interpreted with caution. Lower wastage levels for PCV and Penta have been reported in Africa and Asia [8, 10].

**Table 6. Vaccine stock (physical count) at the facilities on the visit day.**

| Vaccine | Number of facilities affected n(%) | | | | | Min-Max doses in stock |
|---|---|---|---|---|---|---|
| | Out of stock | Expired doses present | Total [a] quantity of expired doses | VVM3/4 doses | Number of facilities with Open unlabeled vials | |
| BCG | 2 (9.1) | 2 (9.1) | 180 | | | 0–2000 |
| Hep B | 5(22.7) | 4 (18.1) | 482 | | 1(4.5) | 0–410 |
| OPV | 2(9.1) | 1 (4.5) | 10 | | 3(13.6) | 0–1720 |
| Penta | 0(0.0) | 0 (0.0) | 0 | | | 1–980 |
| PCV | 2(9.1) | 0 (0.0) | 0 | | | 0–640 |
| MR | 3(13.6) | 1 (4.5) | 50 | | | 0–1155 |
| TT | 2(9.1) | 0 (0.0) | 0 | | 5(22.7) | 0–785 |
| IPV | 4(18.1) | 0 (0.0) | 0 | 32 | | 0–440 |

[a] total in all affected facilities, TT changed to Td in 2018.

Note: HPV which was being rolled out was present in 9 facilities with stock range 0–1630, one facility had no fridge, one had no vaccines in fridge (included as out of stock).

At the central level, only expired doses were reported. However, the inventory was not balanced for almost all the vaccines. We suggest that the m-Supply should incorporate recording the type and number of vaccine doses wasted, and VVM status. Regularly reviewing the data to account for the differences / surpluses will support the aim of strengthening the supply chain systems to ensure uninterrupted supply of vaccines. Challenges with data recording have been documented elsewhere [14].

Expired doses were less commonly reported at the facilities, instead close to half of the wasted doses were reported as 'damaged'. A more detailed description will help particularly because the places are remote and maintaining the cold chain during transportation may be an issue. Power outage was not frequently reported except in the capital, where one facility also reported a faulty refrigerator. About a quarter of the vaccines were discarded based on the VVM indicator change. Solar fridges supplied by UNICEF are available at most service level facilities. Honiara, the capital however does not have (or need) these solar fridges because it has regular mains electricity power supply.

Stockouts were common at all facilities both from the reports and as observed during the survey. Better monitoring will improve the system, in addition to refresher training for health workers and other strategic approaches to effective vaccine management. The number of facilities with no records may reflect the challenge with availability of data. Recording at the national level though not optimal was more complete compared to the facilities. The supply chain needs to be reviewed to ensure there is a clear path between the NMS and the SLMS and other facilities. Findings from this wastage assessment and other EVM recommendations prompted a recent attempt at using participatory action research to address challenges with vaccine management in the Solomon Islands [15]. It will be interesting to see sustainable changes through these approaches and if recording and wastage rates improves over time in the provinces that this study was conducted. Conversely the impact of the COVID-19 pandemic which has had immense disruptions on immunisation services globally [16] poses additional threat to already weak systems. The Solomon Islands are further challenged by climate change. A recent study in the Pacific, showed that >70% (219/303) of the health facilities in Solomon Island were in locations highly susceptible to sea level rise, particularly during high tides and storm surges [17]. Moving forward, strengthening health information systems would be crucial for building resilience.

## Limitations

Several limitations need to be highlighted. The greatest of these is the paucity of data. Since the data were incomplete, it will be difficult to conclude which direction we may have under/over-estimated wastage rates. It was not possible to ascertain a zero-reporting system, hence it is difficult to make any conclusions on the 0% wastage rates at several facilities. Immunisation data where not easily retrievable at the facility hence the decision to collect the information centrally from the EPI program office. The facilities can include wasted doses in their monthly reports to the EPI. They should be encouraged to do zero reporting. Our results on the type of vaccine wastage are limited with close to half of the documented doses simply recorded as wasted or damaged which is not specific enough. To be useful for decision making, more detailed reports on the reason for wastage are needed.

We purposefully selected the provinces and showed very little data from the two remotest provinces of Temotu and Renbel. The other provinces may be different and hence we cannot generalise our results. By selecting the closest and the farthest we have to a large extent provide a good overview of the challenges in the islands and since the three provinces with data account for over 60% of the vaccine consumption, the estimates are likely to be close to the national rates. A prospective data collection through a vaccine wastage sentinel surveillance in each Province maybe needed to better understand the underlying problems and to proffer specific solutions. Despite the limitations, our study gives an indication of wastage for two consecutive years augmented with real time observations at the facilities, which is being used for decision making and could be extrapolated to other small island developing states with similar logistical challenges of small sparse populations over many islands or difficult terrains.

## Conclusion

This vaccine wastage assessment was conducted as part of vaccine management and broader immunization supply chain strengthening in the Solomon Islands, most especially to prepare the country for a sustainable transition from Gavi support. Coupled with findings and recommendations from previously conducted effective vaccine management assessments, the developed vaccine wastage reduction strategy is being implemented to address identified gaps in vaccine handling, data management, temperature recording and supportive supervision among other items to further reduce wastage rates and improve immunization coverage. In addition, the wastage rate estimates are being used to finetune annual nationwide vaccine forecasts and procurements for the country. These will improve the reach, efficiency, and effectiveness of the immunization program. Being the first such wastage assessment conducted in similar contexts of small sparse populations with difficult terrains, it could provide further insights for similar contexts in vaccine wastage rates. Moreover, instituting a vaccine wastage sentinel surveillance could ensure continuous monitoring of vaccine wastage and contribute to program efficiency.

## Supporting information

**S1 Text. Tables A-E.**
(DOCX)

**S1 Data.**
(XLSX)

## Acknowledgments

We thank the data collectors, the staff at the health facilities and the UNICEF administrators for their tremendous support.

## Author Contributions

**Conceptualization:** Ibrahim Dadari, Effua Usuf.

**Data curation:** Ibrahim Dadari, Aven Patson, Jenny Gaiofa, Effua Usuf.

**Formal analysis:** Ibrahim Dadari, Effua Usuf.

**Funding acquisition:** Ibrahim Dadari.

**Investigation:** Ibrahim Dadari, Laura Ropiti, Philip Okia, Salome Namohunu, Divinal Ogaoga, Jenny Gaiofa.

**Methodology:** Ibrahim Dadari, Jenny Gaiofa, Effua Usuf.

**Project administration:** Effua Usuf.

**Supervision:** Ibrahim Dadari, Laura Ropiti, Philip Okia, Jenny Narasia, Timothy Hare'e, Salome Namohunu, Jenny Gaiofa, Effua Usuf.

**Writing – original draft:** Ibrahim Dadari, Effua Usuf.

**Writing – review & editing:** Ibrahim Dadari, Laura Ropiti, Aven Patson, Philip Okia, Jenny Narasia, Timothy Hare'e, Salome Namohunu, Divinal Ogaoga, Jenny Gaiofa, Effua Usuf.

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
