## [Decision Letter · Decision Letter 0]

8 Mar 2022

PGPH-D-21-00311

An assessment of vaccine wastage in the Solomon Islands

Dear Dr. Usuf,

Thank you for submitting your manuscript to PLOS Global Public Health. After careful consideration, we feel that it has merit but does not fully meet PLOS Global Public Health’s publication criteria as it currently stands. Therefore, we invite you to submit a revised version of the manuscript that addresses the points raised during the review process.

We look forward to receiving your revised manuscript.

Kind regards,

Soumyadeep Bhaumik

Academic Editor

Journal Requirements:

1. Please amend your detailed Financial Disclosure statement. This is published with the article, therefore should be completed in full sentences and contain the exact wording you wish to be published.

State what role the funders took in the study. If the funders had no role in your study, please state: “The funders had no role in study design, data collection and analysis, decision to publish, or preparation of the manuscript.”

2. Please update your Competing Interests statement. If you have no competing interests to declare, please state: “The authors have declared that no competing interests exist.”

3. In the online submission form, you indicated that “The data will be shared upon request.”. All PLOS journals now require all data underlying the findings described in their manuscript to be freely available to other researchers, either 1. In a public repository, 2. Within the manuscript itself, or 3. Uploaded as supplementary information.

4. We ask that a manuscript source file is provided at Revision. Please upload your manuscript file as a .doc, .docx, .rtf or .tex. If you are providing a .tex file, please upload it under the item type ‘LaTeX Source File’ and leave your .pdf version as the item type ‘Manuscript’.

5. Please provide separate figure files in .tif or .eps format only and remove any figures embedded in your manuscript file. Please ensure that all files are under our size limit of 20MB.

6. We notice that your supplementary tables are included in the manuscript file. Please remove them and upload them  with the file type 'Supporting Information'. Please ensure that all Supporting Information files are included correctly and that each one has a legend listed in the manuscript after the references list.

Additional Editor Comments (if provided):

The manuscript is within scope of the journal and is of immense significance not only in the geographic area of focus but also overall in the domain - considering the topic at hand. In addition to comments raised by reviewers, authors might need to address following:

Abstract:

In methods section information on how health facilities were selected might be added. Any implication of the sampling on the results might also be highlighted in the conclusion.

Main text:

Please remove funding information from the aim of the project. State the aim only (line 62-64)

The heading of country background might be replaced by setting and merged with the heading of EPI. This can be more concise.

In sampling, please clarify how SLMS were selected? Was it also purposive? Was it based on convenience? This would also feed into discussion and the consequent conclusions derived from the data.

Data sources should precede Data collection section

Though not directly relevant from the data, authors might want to add a few sentences on what the results would mean considering the pandemic and expected impact of climate change in Solomon Islands(data is pre-pandemic).

Reviewers' comments:

Reviewer's Responses to Questions

**Comments to the Author**

1. Does this manuscript meet PLOS Global Public Health’s publication criteria? Is the manuscript technically sound, and do the data support the conclusions? The manuscript must describe methodologically and ethically rigorous research with conclusions that are appropriately drawn based on the data presented.

Reviewer #1: Yes

Reviewer #2: Yes

2. Has the statistical analysis been performed appropriately and rigorously?

Reviewer #1: N/A

Reviewer #2: Yes

3. Have the authors made all data underlying the findings in their manuscript fully available (please refer to the Data Availability Statement at the start of the manuscript PDF file)?

Reviewer #1: Yes

Reviewer #2: Yes

4. Is the manuscript presented in an intelligible fashion and written in standard English?

Reviewer #1: Yes

Reviewer #2: Yes

5. Review Comments to the Author

Reviewer #1: This is an important study for economic and epidemiological reasons. It highlights difficulties in efficiently maintaining stocks of EPI vaccines, and making them widely available to individuals even though many vaccines are provided in multi-dose vials which can expire. Overall, the study was well conducted and the discussion raises recommendations in line with the study results.

I feel like an additional point to consider from your paper would be to increase vaccine durability. Has much research been done on re-usability of multi-dose vials, etc.?

Line 72: could you check the birth cohort number?

Can you mention what is a single dose and multi dose vial vaccine in the methods?

As is, Tables 3 and 4 are okay, but they contain a lot of information. I would maybe move that to a supplementary appendix, and then just summarize the ranges of numbers within a table in the main text.

Could you explain somewhere what a negative number would mean? (perhaps a limitation of problems with bookkeeping, or getting vaccines transfered between clinics?)

Could you define/ write out VVM?

In the discussion you mention an "acceptable range" for wastage rates. Could you define what that would be when you first mention it (paragraph 4 you start to get into this - but should move this material forward, or even add to introduction).

Reviewer #2: This manuscript presents the vaccine wastage in Solomon Islands. The manuscript gives important information about the vaccine logistics management in Solomon Islands. The manuscript can be improved further if the following points are addressed.

1. Introduction

Should mention, if there is any practice of monitoring vaccine wastage periodicals as part of the program or it is done as an adhoc manner like this project.

2. Methods- should mention

What are the types of cold chain devices used for storage at the different levels and their power supply type? How the program covers the islands for service delivery? How the vaccines are transported between these levels of stores at at what frequency?

in data collection: better to specify the months: Jan-Dec for the years

3. Results

3.1. Fig 2: It shall be better to give the number for Guadacanal as others.

3.2. Table 1: What does +ve and -ve indicate: please clarify in foot note.

Can mention the damaged doses also here as a column.

3.3. Table 2: It may be better to add the number of vaccines received/supplied also.

3.4. Overall wastage rates

How the 100% wastage rates were explained- no session held or all vials were damaged?

3.5. Types of wastage

Were the differences in stock consistent across the years and for all types of vaccines or periodic observations?

3.6. Fig 3:

What does the VVM mean- due to VVM expiry?

what dose wastage mean- date expiry?

What does the wastage mean- due to opened vial wastage?

3.7. Narration for Table 5

How were the vaccines stored in absence of the refrigerators? Were the VVM related wastage could be linked to the periods of no refrigerator?

3.8. Vaccines - Physical count at the facilities

What was the level or agreement and/or differences between the stock numbers from the record/register and physical count?

3.9. Table 5

Section- Vaccines returned to cold chain

Which vaccines- used, unused or partially used? - need to clarify

4. Discussion

4.1. Should discuss the potential reasons and factors influencing the high wastage in some islands and for some vaccines than others. Should discuss the vaccine record keeping status and need for improvement with appropriate documentation for wastage estimation.

4.2. Limitations should mention the lack of information about the type of wastage in some islands.

5. Supplementary table 2

Can add population size also in the table.

6. PLOS authors have the option to publish the peer review history of their article (what does this mean?). If published, this will include your full peer review and any attached files.

**Do you want your identity to be public for this peer review?** For information about this choice, including consent withdrawal, please see our Privacy Policy.

Reviewer #1: No

Reviewer #2: **Yes: **Manoja Kumar Das

---

## [Decision Letter · Decision Letter 1]

11 May 2022

An assessment of vaccine wastage in the Solomon Islands

PGPH-D-21-00311R1

Dear Dr Usuf,

We are pleased to inform you that your manuscript 'An assessment of vaccine wastage in the Solomon Islands' has been provisionally accepted for publication in PLOS Global Public Health.

Best regards,

Soumyadeep Bhaumik

Academic Editor

Reviewer Comments (if any, and for reference):

Reviewer's Responses to Questions

**Comments to the Author**

1. If the authors have adequately addressed your comments raised in a previous round of review and you feel that this manuscript is now acceptable for publication, you may indicate that here to bypass the “Comments to the Author” section, enter your conflict of interest statement in the “Confidential to Editor” section, and submit your "Accept" recommendation.

Reviewer #1: All comments have been addressed

Reviewer #2: All comments have been addressed

2. Does this manuscript meet PLOS Global Public Health’s publication criteria? Is the manuscript technically sound, and do the data support the conclusions? The manuscript must describe methodologically and ethically rigorous research with conclusions that are appropriately drawn based on the data presented.

Reviewer #1: Yes

Reviewer #2: Yes

3. Has the statistical analysis been performed appropriately and rigorously?

Reviewer #1: Yes

Reviewer #2: Yes

4. Have the authors made all data underlying the findings in their manuscript fully available (please refer to the Data Availability Statement at the start of the manuscript PDF file)?

Reviewer #1: Yes

Reviewer #2: Yes

5. Is the manuscript presented in an intelligible fashion and written in standard English?

Reviewer #1: Yes

Reviewer #2: Yes

6. Review Comments to the Author

Reviewer #1: I believe the authors have comprehensively responded to all previous comments.

Reviewer #2: Authors have addressed the comments raised during earlier review.

7. PLOS authors have the option to publish the peer review history of their article (what does this mean?). If published, this will include your full peer review and any attached files.

**Do you want your identity to be public for this peer review?** For information about this choice, including consent withdrawal, please see our Privacy Policy.

Reviewer #1: No

Reviewer #2: **Yes: **Manoja Kumar Das
